# Identification of VRK1 as a New Neuroblastoma Tumor Progression Marker Regulating Cell Proliferation

**DOI:** 10.3390/cancers12113465

**Published:** 2020-11-20

**Authors:** Ana Colmenero-Repiso, María A. Gómez-Muñoz, Ismael Rodríguez-Prieto, Aida Amador-Álvarez, Kai-Oliver Henrich, Diego Pascual-Vaca, Konstantin Okonechnikov, Eloy Rivas, Frank Westermann, Ricardo Pardal, Francisco M. Vega

**Affiliations:** 1Instituto de Biomedicina de Sevilla (IBiS), Hospital Universitario Virgen del Rocío/CSIC/Universidad de Sevilla, 41013 Seville, Spain; acolmenero-ibis@us.es (A.C.-R.); magomez-ibis@us.es (M.A.G.-M.); ismaelropri@us.es (I.R.-P.); aaalvarez@us.es (A.A.-Á.); 2Departamento de Fisiología Médica y Biofísica, Universidad de Sevilla, 41013 Seville, Spain; 3Departamento de Biología Celular, Facultad de Biología, Universidad de Sevilla, 41012 Seville, Spain; 4German Cancer Research Center (DKFZ), Division Neuroblastoma Genomics, 69120 Heidelberg, Germany; k.henrich@kitz-heidelberg.de (K.-O.H.); f.westermann@kitz-heidelberg.de (F.W.); 5Departamento de Anatomía Patológica, Hospital Universitario Virgen del Rocío, 41013 Sevilla, Spain; diego.pascualvaca.sspa@juntadeandalucia.es (D.P.-V.); eloy.rivas.sspa@juntadeandalucia.es (E.R.); 6Pediatric Neurooncology, German Cancer Research Center (DKFZ), 69120 Heidelberg, Germany; k.okonechnikov@kitz-heidelberg.de

**Keywords:** neuroblastoma, high-risk, VRK1, proliferation, MYCN, tumorigenesis

## Abstract

**Simple Summary:**

Aggressive neuroblastoma (NB) is one of the most common pediatric cancers and causes a disproportionate mortality among affected children. A better knowledge about the biology of this tumor is needed to be able to provide new treatments and prognostic tools. Protein kinases are one of the best targets for molecular cancer treatment, as we are potentially able to produce inhibitors that abrogate its activity. In this study we have identified that the human protein kinase VRK1 is associated with tumor aggressiveness and patient survival in NB. We have characterized the function of VRK1 in NB tumor cells and determined that VRK1 is an essential mediator of NB cell proliferation. We also study the relationship between VRK1 and the oncogene MYCN, the best-known marker for NB progression to date. Our work suggests that VRK1 synergize with MYCN to drive NB progression and that VRK1 inhibition may constitute a novel cell-cycle-targeted strategy for anticancer therapy in neuroblastoma.

**Abstract:**

Neuroblastoma (NB) is one of the most common pediatric cancers and presents a poor survival rate in affected children. Current pretreatment risk assessment relies on a few known molecular parameters, like the amplification of the oncogene MYCN. However, a better molecular knowledge about the aggressive progression of the disease is needed to provide new therapeutical targets and prognostic markers and to improve patients’ outcomes. The human protein kinase VRK1 phosphorylates various signaling molecules and transcription factors to regulate cell cycle progression and other processes in physiological and pathological situations. Using neuroblastoma tumor expression data, tissue microarrays from fresh human samples and patient-derived xenografts (PDXs), we have determined that VRK1 kinase expression stratifies patients according to tumor aggressiveness and survival, allowing the identification of patients with worse outcome among intermediate risk. VRK1 associates with cell cycle signaling pathways in NB and its downregulation abrogates cell proliferation in vitro and in vivo. Through the analysis of ChIP-seq and methylation data from NB tumors, we show that VRK1 is a MYCN gene target, however VRK1 correlates with NB aggressiveness independently of MYCN gene amplification, synergizing with the oncogene to drive NB progression. Our study also suggests that VRK1 inhibition may constitute a novel cell-cycle-targeted strategy for anticancer therapy in neuroblastoma.

## 1. Introduction

Neuroblastoma is a pediatric solid tumor with embryonic origin derived from sympathoadrenal precursors of the neural crest [1]. It is the most common type of cancer diagnosed during the first year of life and the most frequent extracranial solid tumor in children [2]. Neuroblastoma is clinically characterized by a great heterogeneity, with patients presenting extensive metastasis and with frequent relapses. This cancer displays an event-free survival below 50% [3]. At the cellular level, intratumoral heterogeneity has emerged as a hallmark for these tumors, with presence of cell populations that differ in their differentiation status, proliferative potential and response to treatment [4,5]. Aggressive neuroblastomas are incurable to date, reflecting the pressing need for a better understanding of the cellular and molecular mechanisms that mediate aggressiveness, relapse and metastasis, in order to elucidate new and better prognostic markers and therapeutic targets.

Although neuroblastomas can occur in familial contexts, most cases arise sporadically. Few recurrent alterations in common oncogenes or tumor suppressors have been identified. Among those, the oncogene MYCN is amplified in about 20–30% of all tumors, being associated with poor prognosis [6]. Many aggressive tumors however do not harbor MYCN amplification [7]. It is believed that many of the changes leading to neuroblastoma initiation and progression are aberrant epigenetic events affecting transcriptional programs, probably linked to processes occurring during sympathoadrenal development from the neural crest [8,9].

Vaccinia-related kinase 1 (VRK1) is a member of a ser/thr kinase family that phosphorylates various molecules and transcription factors implicated in chromatin condensation, DNA repair and cell cycle progression [10,11,12]. VRK1 acts in some context as a chromatin remodelling enzyme, as it is known to phosphorylate several histones, affecting their acetylation and methylation status, and influencing gene transcription, DNA damage response and cell cycle [11,13]. A physiological role for VRK1 has been described in fetal tissues, for example during uterine development or embryonic development of the hematopoietic system [14,15]. VRK1 is also highly expressed in spermatogonia stem cells, being essential for spermatogonia cell maintenance [16].

VRK1 activity has been also associated with pathological situations. Genetic variants of VRK1 have been linked to diverse neurodegenerative disorders [17,18]. High VRK1 expression has been associated with poor prognosis in various cancers like head and neck [19], lung carcinomas [20] or hepatocellular carcinomas [21,22]. Furthermore, it is known that high VRK1 protein levels confer a stronger resistance to treatment in breast [23] and lung cancer [24]. The possible implication of VRK1 in neuroblastoma or other pediatric cancers is unknown. However, the *VRK1* gene has been suggested as a potential transcriptional target for the oncoprotein MYCN [25].

Given the evidence linking VRK1 to tumor progression and the connection with MYCN, we decided to analyze the contribution of this protein to neuroblastoma cell biology, focusing on its pathological significance and prognostic value. We demonstrate that VRK1 is highly expressed in high-grade neuroblastoma and is associated with proliferation and dedifferentiation in tumor cells. Functional experiments show that VRK1 is essential for NB cell proliferation and tumor progression, and thus could be a new target for neuroblastoma treatment. Despite being a target of MYCN transcription factor, VRK1 is a marker of tumor progression and malignancy independent of MYCN expression. The expression of VRK1 can serve as a prognostic factor for MYCN nonamplified tumors with malignant progression or to stratify intermediate grade patients with uncertain outcome.

## 2. Results

### 2.1. VRK1 Expression Correlates with Aggressiveness in Neuroblastoma Tumors

To elucidate the possible implication of VRK1 in NB tumors, we explored the expression of *VRK1* in several cohorts of human NB patient tumor samples, using expression data from public databases. *VRK1* is highly expressed in the International Neuroblastoma Staging System (INSS) stage 4 NB compared with normal adrenal tissue, and significant differences can be observed between more aggressive neuroblastoma stages (stages 3 and 4) and more benign ones (Stages 1, 2 and 4S) (Figure 1a and Appendix A).

High VRK1 expression levels were also able to significantly stratify patients with high-risk, unfavorable histology, poor survival and worse outcome (Figure 1b–d and Appendix A), indicating a strong correlation between VRK1 expression and NB unfavorable prognosis and aggressiveness. Same results were obtained with all patient datasets tested. A proportion of patients classified at diagnosis as INSS Stage 3 present progression of the disease later on. Strategies to stratify these patients early on would be beneficial to anticipate treatment. VRK1 expression alone could identify, among INSS stage 3 patients, those with high-risk, unfavorable histology, worse outcome and low survival probability, indicating the potential of VRK1 expression as a prognostic factor (Figure 1e–h). High VRK1 expression is also indicative of worse survival probability among the patients with stage 3 tumors without MYCN amplification and among low- and intermediate-risk patients (Figure 1i,j). VRK1 expression also identifies patients with worse survival when classified into age at diagnosis groups, including groups in which it would be important to find new prognostic markers to identify patients with tumor progression, such as the ones diagnosed before 12 months of age (Appendix A).

In addition to mRNA expression, VRK1 protein detection by immunohistochemistry on our own collection of 36 human neuroblastoma tumor samples shows high VRK1 expression in M stage tumors and patient-derived xenografts (PDXs) derived from high-grade NB patients, compared to L1 tumors (Figure 2a). Interestingly, we observed a decrease in VRK1 expression in samples obtained after treatment, when compared to the ones obtained at diagnosis, but high VRK1 expression on samples from patients with relapses and worse outcome (Appendix A).

We finally analyzed VRK1 protein levels in a panel of commonly used neuroblastoma cell lines and primary PDX-derived cells. VRK1 expression was variable among the cell lines and does not seem to correlate with MYCN amplification, but the kinase was expressed in all neuroblastoma cell lines and PDX samples tested (Figure 2b,c). We decided to perform functional studies in SK-N-SH and IMR-32 cell lines, with similar levels of VRK1, a moderate expression and different MYCN-amplification status.

### 2.2. VRK1 is Associated with NB Tumor Cell Proliferation

A role for VRK1 in the control of cell division has been proposed in cells from diverse origin [12]. To analyze the specific implication of VRK1 in neuroblastoma tumors, we first took an indirect approach in which, using different human neuroblastoma expression datasets, we analyzed the function of genes whose expression were significantly correlated to the one of *VRK1* across the samples, creating a *VRK1*-high-expression signature (Figure 3a and Appendix A). A signaling pathway analysis performed with these genes showed a significant enrichment in pathways related to cell cycle, DNA replication, DNA repair and differentiation. There was a strong positive correlation in NB tumors and PDX-derived cells between *VRK1* expression and the expression of the proliferation marker *Ki67*, as well as the mitotic index in the tumors, observed both by mRNA and immunohistochemistry (Figure 3b–d).

VRK1 is an essential gene in many cellular contexts, and a total abrogation of VRK1 expression has been shown to be detrimental for cells [15,26]. Therefore, we used transient siRNA transfection to partially reduce the expression of VRK1 in neuroblastoma cells and study the functional consequences. VRK1 downregulation in NB cells by specific siRNA diminishes cell division, observed as a significant reduction in cell culture confluency, cell viability and cycling Ki67-positive cells (Figure 3e,f). Competition assays performed with SK-N-SH cells labeled either with Green Fluorescent Protein (GFP) or Red fluorescent Protein (RFP), and depleting VRK1 with siRNA only in one of the two populations, showed an enrichment in cells transfected with control siRNA after 96 h in culture (Appendix A). Single-cell clonogenic proliferation assays also showed an NB cell proliferation dependence on VRK1 (Appendix A). Cell cycle and apoptosis analysis by FACS showed that cells depleted of VRK1 do not experience an increase in apoptosis (Appendix A).

All together, these functional assays demonstrate an important role for VRK1 in NB cell proliferation. Concordant with this, transient and moderate VRK1 knockdown induces a downregulation of cell cycle progression protein levels, such as cyclin D1 or mdm2, and an increase in cell cycle suppressors like *p* 53 and its target *p* 21 (Figure 3g). Interestingly, the downregulation of VRK1 also provokes a drop in MYCN protein expression.

### 2.3. VRK1 Downregulation Impairs Neuroblastoma Tumorigenesis in a Xenograft Model

Given the association of VRK1 with malignancy, we investigated whether VRK1 depletion on neuroblastoma cells would impact tumor progression. Neuroblastoma cells were treated with siRNA against VRK1 or with control siRNA before xenograft transplantation on recipient immunocompromised mice (Figure 4). Tumor onset was delayed on mice injected with VRK1-depleted cells, and resulting tumors in these mice were significantly smaller. A histological analysis showed that tumors grown from VRK1-RNAi-treated cells were less proliferative. Surprisingly, depletion of VRK1 with siRNA is transitory and tumors were collected after 8 weeks, indicating that VRK1 downregulation is somehow maintained in tumors and might have a profound and long-lasting influence on tumor establishment and growth.

### 2.4. VRK1 Associates with Neuroblastoma Progression Independently of MYCN Amplification

We have previously seen that *VRK1* expression strongly correlates with aggressiveness in neuroblastoma tumors. *VRK1* expression is also significantly higher in *MYCN*-amplified versus nonamplified neuroblastoma tumors (Figure 5a). However, the correlation of *VRK1* expression with malignant traits was still maintained when only the *MYCN* nonamplified tumors were analyzed (Figure 5b). This indicates that, despite the possible relationship between *VRK1* and *MYCN*, *VRK1* expression strongly correlates with neuroblastoma unfavorable prognosis and aggressiveness independently of the *MYCN* amplification status.

We decided to further explore the connection between *MYCN* transcription factor and VRK1 in neuroblastoma. Amplification of *MYCN* in neuroblastoma tumors correlates with high *VRK1* gene expression and aggressiveness, although the levels of *VRK1* do not correlate with the levels of *MYCN* expression in nonamplified tumors (Figure 5c). In neuroblastoma PDX tissues, all MYCN-positive cells showed expression of VRK1 (Figure 5d). These could be explained because *VRK1* has been identified as a MYCN transcriptional target with an E-box MYCN response element in the gene promoter [25]. Analysis of *VRK1* gene promoter methylation status in an NB tumor panel shows a region, downstream of the *VRK1* transcription start site, that is hypomethylated in high-risk tumors with *MYCN* amplification and is defined by a single CpG probe (cg26685539, mean methylation difference 0.2 (*b*), *q* < 0.001)) (Figure 6a, Appendix A). According to MYCN ChIP-seq data from three neuroblastoma cell lines (Be(2)-C, Kelly and NGP), this region does not map to the uniformly hypomethylated *MYCN*-binding site, and possibly indicates the binding of additional transcription factors or coactivators. Significant negative correlation between cg26685539 methylation and VRK1 expression indicates the presence of a promoter downstream correlated region (pdCR) [27] with high potential for *VRK1* gene transcription regulation (Figure 6b). This opens the possibility to synergistic actions regulating cell proliferation in neuroblastoma cells. *MYCN*-amplified neuroblastoma cell lines are more sensitive to VRK1 inhibition than cells with no *MYCN* amplification, reinforcing this possibility (Figure 6c).

## 3. Discussion

Recurrent mutations or alterations in known oncogenes are not common in neuroblastoma. High *MYCN* expression occurs in about 20% of neuroblastoma and is a clear indication of bad prognosis. *MYCN* gene amplification confers a high probability of aggressive disease and metastasis [28]. However, there is still a proportion of NBs with poor survival but no clear molecular prognosis indicator at diagnosis or therapeutic target. We have shown that VRK1 kinase is highly expressed in NBs of high grade and poor survival, providing a potential new gene target with therapeutic value.

Our gene expression and ontology analysis predicts that *VRK1* expression is associated in neuroblastoma tumors with signaling pathways involved in cell cycle regulation, DNA replication or DNA repair. VRK1 has been described as a proliferation control protein in human fibroblasts and in some cancers like head and neck or myeloma [19,29,30]. We have identified that VRK1 is associated with cell proliferation in NB tumor cells and that it is an essential gene controlling cell division in these tumors. *VRK1* expression downregulation associates with cell cycle arrest, observed by a reduction of cyclin D1 and MDM2 and an increase of p 53 and p 21, while no increase in apoptosis was observed. These indicate a possible cancer type-dependent action of VRK1, as higher apoptotic activity was detected in esophageal squamous cell carcinoma cells after VRK1 downregulation [22]. We found that VRK1 expression in NB also associates with genes belonging to signaling pathways related to differentiation. Given the importance of sympatoadrenal differentiation for the origin and development of NB, it would be interesting to study a possible role of VRK1 in this phenomenon.

A complete abrogation of VRK1 expression has been shown to be deleterious in cells and during development [15,18]. These have posed difficulties for the generation of knockout mice models or the use of CRISPR-Cas9 to study VRK1 biology. Surprisingly, a transient and partial reduction of VRK1 expression in NB cells was enough to significantly impaired tumor growth in a xenograft mouse model. Moreover, a sustained reduction in VRK1 expression in tumor cells in vivo could be observed long after transfection. It is possible that VRK1 downregulation elicits a long-lasting epigenetic effect on NB tumor cells or that a high VRK1 expression is needed for the successful establishment of the tumor in the initial steps. VRK1 is known to induce epigenetic changes in chromatin by histone phosphorylation and acetylation modulation [23,31]. We cannot rule out an influence of the tumor microenvironment on VRK1 expression and the proliferative potential of NB cells, not triggered in the case of tumor establishment with VRK1 knockdown cells. In any case, this could place VRK1 as an essential gene for NB progression, suggesting a possible use of transient inhibition of VRK1 for the treatment of NB. So far, we lack compounds to specifically inhibit the VRK family of proteins, but different efforts are underway to produce potent and specific inhibitors [11,32].

Interestingly, *VRK1* expression alone stratified patients originally diagnosed as intermediate risk or INSS stage 3, allowing the identification of patients with worse outcome. Although most stage 3 patients respond well to therapy and become disease free, about 10-15% of them do not respond adequately to current treatments, many of them but not all, corresponding to patients with *MYCN* amplified tumors. *VRK1* expression, together with known prognostic factors, might contribute to a better and more precise stratification of intermediate risk patients. Further studies with bigger cohorts, especially in the case of stage 3 *MYCN* non-amplified tumors, could finally help to determine the potential use of VRK1 expression in the clinic.

VRK1 expression significantly correlates with malignancy in NB patient tumors, even in tumors without MYCN amplification, suggesting that VRK1 might be an indicative of poor survival, independent of MYCN status. We have observed a significantly higher expression of VRK1 gene in MYCN amplified tumors, although the correlation with MYCN gene expression disappears in tumors without amplification. The VRK1 gene promoter contains an E-BOX MYCN response sequence, and ChipSeq data from NB cells confirms MYCN binding to the VRK1 gene. Interestingly, VRK1 downregulation seems to also affect MYCN expression, raising the possibility of a regulatory feedback loop, similar to the one occurring between VRK1 and other regulators, like *p* 53 [33]. Both VRK1 and MYCN proteins could be cooperating in NB progression, although VRK1 gene expression does not seem to be entirely dependent on MYCN transcriptional activation. Interestingly, high levels of MYCN in MYCN-amplified tumors are associated with a specific epigenetic landscape in the VRK1 gene, characterized by hypomethylation of a region downstream of the promoter, which correlates with high expression of the kinase. This opens up the possibility for the presence of a new unknown transcription factor collaborating with MYCN to further activate VRK1 in MYCN-amplified NB tumors. MYCN-amplified tumor cells seem to be dependent on VRK1 expression for its exacerbated proliferation, raising the possibility of using VRK1 inhibitors for NB treatment as an alternative to MYCN targeting. Furthermore, the use of VRK1 inhibitors in combination with BET bromodomain domain inhibitors, with action against MYCN transcriptional activity, might also be a promising option [34,35].

## 4. Materials and Methods

### 4.1. Human Tumor Gene Expression and Methylation Analysis

*VRK1* gene expression analysis was performed with the use of the R2: Genomics Analysis and Visualization Platform (Available on: http://r2.amc.nl; Academic Medical Centre, Amsterdam) and GEO Datasets (Gene Expression Omnibus; NCBI: National Centre for Biotechnology Information, Bethesda, MD, USA). Patient cohort datasets used are summarized in Appendix A. Tumors on these datasets are clinically classified following the International Neuroblastoma Staging System (INSS) [36], classifying tumors in growing degree of aggressiveness into stages 1, 2, 3 or 4. 4S is a special metastatic benign stage. Risk information on these patient samples according to the INRG stratification system is also used [37]. Gene ontology and pathway analysis were performed with the help of the Reactome Pathway Database (Available on: https://reactome.org).

*VRK1* gene promoter region methylation analysis by Illumina HumanMethylation450 BeadChip was performed with data from [38]. Differential DNA methylation was performed applying the dmpFinder function of the minfi R package with limits *q*-val < 0.05 and mean methylation difference > 0.1 (beta value). MYCN ChIP-seq data from neuroblastoma cell lines was derived from GEO ID GSE80154 [39].

### 4.2. Cell Culture and siRNA Transfection

The human NB cell lines SK-N-SH, SK-N-DZ and IMR-32 were obtained from the EACC (Salisbury, UK) and grown at 37 °C in 5% CO_2_ in Dulbecco’s Modified Eagle Medium (GIBCO) + 10% fetal bovine serum (FBS). CHLA20 and CHLA-255 NB cell lines were obtained from the COG Cell Line and Xenograft Repository (Texas Tech University Health Sciences Center, Lubbock, TX, USA) and were grown in IMDM media (GIBCO) + 15%FBS and 1 × ITS (5 μg/mL insulin, 5 μg/mL transferrin y 5 ng/mL selenic acid). All media was supplemented with 2 mM glutamine (GIBCO BRL, Gaithersburg, MD, USA)_,_ 100 U/mL of penicillin and 100 μg/mL of streptomycin. Cell lines expressing fluorescent proteins were generated by transfection and selection with the corresponding vectors (pEGFPN1 and pERFPN1 from Clontech, Mountain View, CA, USA). NB39T and NB48T PDX primary cell lines were derived from freshly obtained tumors and have been described previously [5].

Two siRNAs targeting VRK1 (named siVRK1(02) and siVRK1(03), respectively) and a control siRNAs (named siControl) were obtained from Dharmacon (Thermo Scientific, Waltham, MA, USA). Transient transfections were performed using Lipofectamine 2000 (Invitrogen, Carlsbad, CA, USA) according to the manufacturer’s instructions with final siRNAs concentration of 100 nM. Knock down of the protein was checked and assays performed 72 h post-transfection, unless indicated otherwise. The sequences of VRK1 siRNAs and control siRNAs were: siVRK1(02) (CAAGGAACCUGGUGUUGAA; UUCAACACCAGGUUCCUUG), siVRK1(03) (GGAAUGGAAAGUAGGAUUA; UAAUCCUACUUUCCAUUCC), siControl-ontarget plus#4 (UGGUUUACAUGUUUUCCUA; UAGGAAAACAUGUAAACCA).

For NB cell line sensitivity to VRK1 knockdown, data from the DRIVE data portal was collected [40].

### 4.3. Western Blot and Cellular Assays

Western blot analysis was performed from cell lysates using the following antibodies: mouse anti-VRK1 (1F6) (1:1000, #3307, Cell Signaling Technology, Danvers, MA, USA), mouse anti-Cyclin D1 (1:500, Santa Cruz Biotechnology, Santa Cruz, CA, USA), rabbit anti-MDM2 (1:500, R&D Systems), rabbit anti-*p* 21 (1:500, Abcam), mouse anti-Nestin (1:500, R&D Systems), mouse anti-MYCN (1:500, Millipore, Burlington, MA, USA), mouse anti-α-Tubulin (1:5000, Sigma) and rabbit anti-GAPDH (1:5000, Trevigen, Gaithersburg, MD, USA).

Cell viability was measured with the use of AlamarBlue reagent (Thermo Scientific, Waltham, MA, USA) 96 h post-transfection, following recommended instructions.

For clone formation assays, cells were transfected with siRNA as described and after 72 h 5 × 10^4^ cells per well were seeded on a 6-well plate and grown for 7 days before pictures of the cell clones formed were taken under the microscope.

Apoptosis assay was performed with the Phycoerythrin (PE) AnnexinV Apoptosis Detection kit (BD Pharmigen, San Diego, CA, USA). The original Western blot figures can be found in the Appendix A (Appendix A).

### 4.4. Tumor Xenografts

6–8 weeks-old CB-17 SCID mice were acquired from Harlan Laboratories. Each mouse was injected in the right flank with 4 × 10^5^ SK-N-SH, 24 h after transfection with the corresponding siRNAs. Efficiency of knock-down was demonstrated in parallel. After 8 weeks, all mice were sacrificed humanely and tumors excised and measured. Tissue samples were included in paraffin for immunohistochemical analysis.

The generation and maintenance of PDXs was described previously [5]. Four different generated PDXs were used (Appendix A). PDX tumor samples were obtained and treated similarly than xenograft.

### 4.5. Immunofluorescence and Immunohistochemistry

For immunofluorescence, cells or frozen sections were fixed with 4% paraformaldehyde, permeabilized with 0.2% Triton X-100 in PBS, blocked with 1% BSA in PBS and incubated with the corresponding antibodies. The following antibodies were used: rabbit anti-VRK1 (1:500, Sigma, San Luis, MO, USA), rabbit anti-Ki67 (1:500, Thermo Scientific, Waltham, MA, USA), mouse anti-MYCN (1:500, Millipore, Burlington, MA, USA). Nuclei were stained with DAPI (Life Technologies, Carlsbad, CA, USA; 1:1000). Secondary antibodies used were Alexa488-donkey-anti-mouse-IgG and Alexa568-goat-anti-rabbit-IgG (Life Technologies, Carlsbad, CA, USA; 1:1000). Images were obtained with an epifluorescence microscope (Olympus BX-61, IX-71 or Nikon TiE2000). Images were analyzed with ImageJ software (National Institutes of Health, Bethesda, WA, USA) and Cell Profiler [41].

Immunohistochemical detection in patient samples was performed using a cohort of 31 primary paraffin-embedded neuroblastoma cancer tissue sections, placed in duplicate on a tissue microarray (TMA) (Appendix A) or PDXs. Tumors on the TMA were clinically classified into the more recent International Neuroblastoma Risk Groups staging system (INRGSS), dividing patients into L1, L2, M or MS stages [42]. For immunohistochemistry staining, paraffin-embedded sections were obtained from tissues, hydrated, treated for antigen retrieval with citrate buffer (pH 6) and incubated with the corresponding antibodies. Vectastain ABC kit and DAB peroxidase substrate kit (Vector Laboratories, Burlingame, CA, USA) were used following recommended procedures. Consecutive sections were used when appropriate. Antibodies used were rabbit anti VRK1 (Sigma, San Luis, MO, USA; 1:500) and rabbit anti Ki67 (Thermo Scientific, Waltham, MA, USA; 1:500). Secondary antibodies used were biotinylated-conjugated anti-mouse and biotinylated-conjugated anti-rabbit (Vector Laboratories Burlingame, CA, USA; 1:1000). Pathologists scored the samples using a scale that combines the percentage of positive cells and the intensity of the reaction product [43].

### 4.6. Statistical Analysis

Statistical analysis was performed on R2 genomics platform or Graphpad Prism software. One-way (two-way on grouped differentiation data) analysis of variance ANOVA or the unpaired *t*-test were used for statistical analysis. Results were assumed significant when *p* < 0.05. Fisher’s exact test was used for the analysis of positive and negative tumor samples. In gene expression analysis plots, box and whiskers graphs show median, 10–90 percentiles and outliers as dots. Scan was used to identify Kaplan survival curves cutoff, and p-values are Bonferroni corrected. Bar graphs show average ± SEM in all cases. Number of measurements or samples (n) is indicated in each case.

### 4.7. Ethics Approval and Consent to Participate

All animal experiments were conducted according to procedures approved by the Ethics Committee from the University of Seville (CEEA_US2014-012) and complying with all animal use guidelines. Fresh neuroblastoma tumor samples were managed and obtained from the Andalusian tissue Biobank (Andalusian Public Health System Biobank and ISCIII-Red de Biobancos PT13/0010/0056, Seville, Spain) after informed consent was obtained from subjects’ guardians following all established regulations. Clinical information on patient cohorts is available for research use and has been anonymized.

## 5. Conclusions

Overall, our findings identify VRK1 as an essential protein for NB cell proliferation, and establish a complex relationship and possible collaboration with the MYCN oncogene driving NB progression.

## Figures and Tables

**Figure 1 cancers-12-03465-f001:**
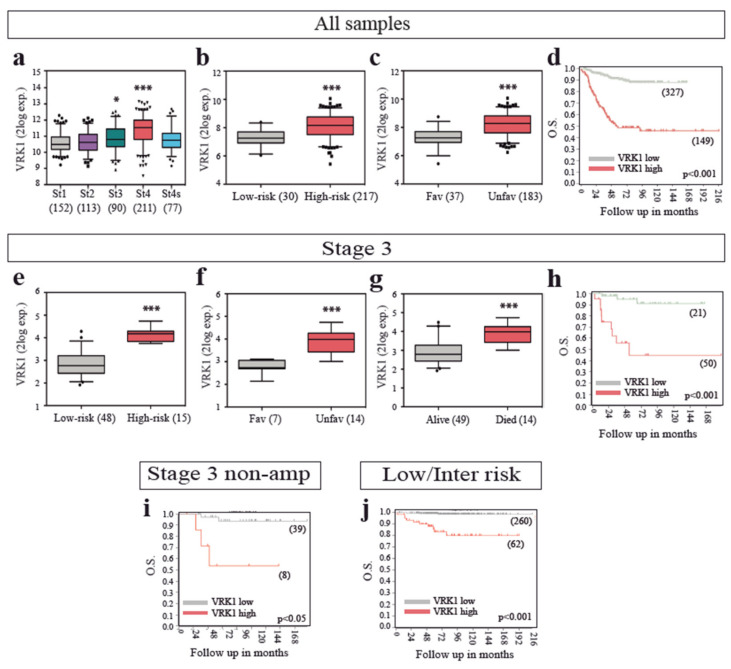
VRK1 mRNA expression correlates with neuroblastoma (NB) progression in patient tumor samples. (**a**–**c**) VRK1 expression in patient tumor samples of all stages separated by the International Neuroblastoma Staging System (INSS) stage a, the International Neuroblastoma Risk Group Staging System (INRGSS) risk b or histology c. GSE62564 and GSE3446 datasets used; (**d**) Kaplan curve showing overall survival probability from patients stratified according to VRK1 expression (GSE62564); (**e**–**g**) stage 3 patient tumor samples separated by risk (**e**) histology (**f**) or outcome (**g**). GSE62564 and GSE3446 datasets used; (**h**–**j**) Kaplan curve showing overall survival probability from stage 3 patients (**h**) stage 3 MYCN nonamplified patients (**i**) or low/intermediate risk patients (**j**) stratified according to VRK1 expression (GSE62564). Number of samples shown in brackets. *: *p* < 0.05; ***: *p* < 0.001.

**Figure 2 cancers-12-03465-f002:**
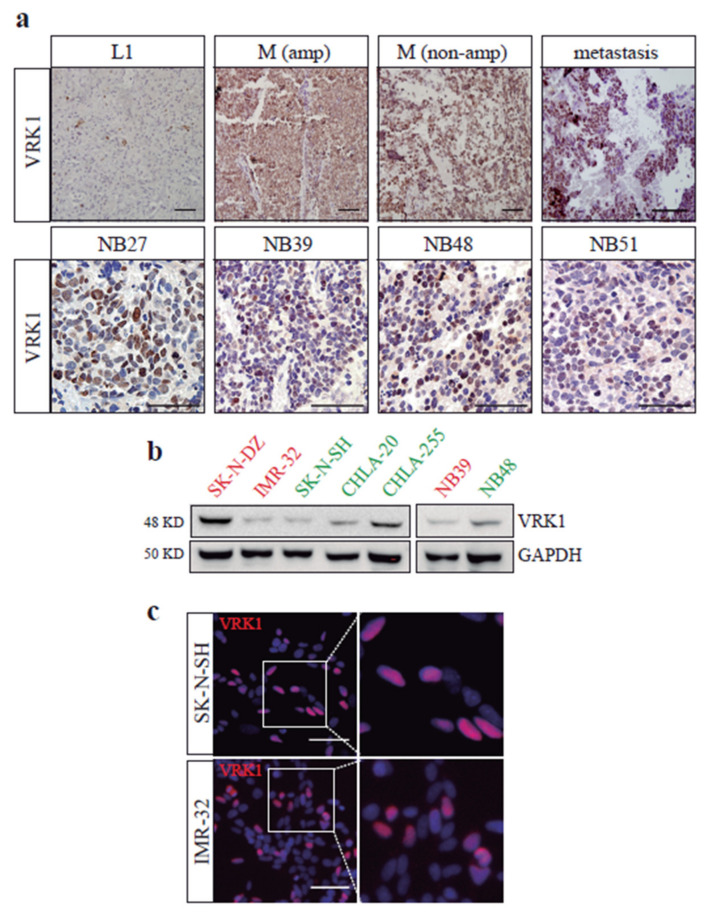
VRK1 protein expression in NB samples and cells. (**a**) Immunohistochemistry showing VRK1 expression in examples of NB tumor patient samples from the indicated International Neuroblastoma Risk Group Staging System (INRGSS) stages (top row) or patient-derived xenografts (PDXs) samples (bottom row), from our in-house collection. Scale bars: 60 μM. (**b**) Western blot showing VRK1 expression in cell lysates from different NB cell lines or PDX-derived cells. In red, MYCN-amplified cell lines. In green, nonamplified cell lines. (**c**) Immunofluorescence showing VRK1 expression (red) in NB cell lines. Nuclear staining is shown in blue. Scale bar: 60 μM.

**Figure 3 cancers-12-03465-f003:**
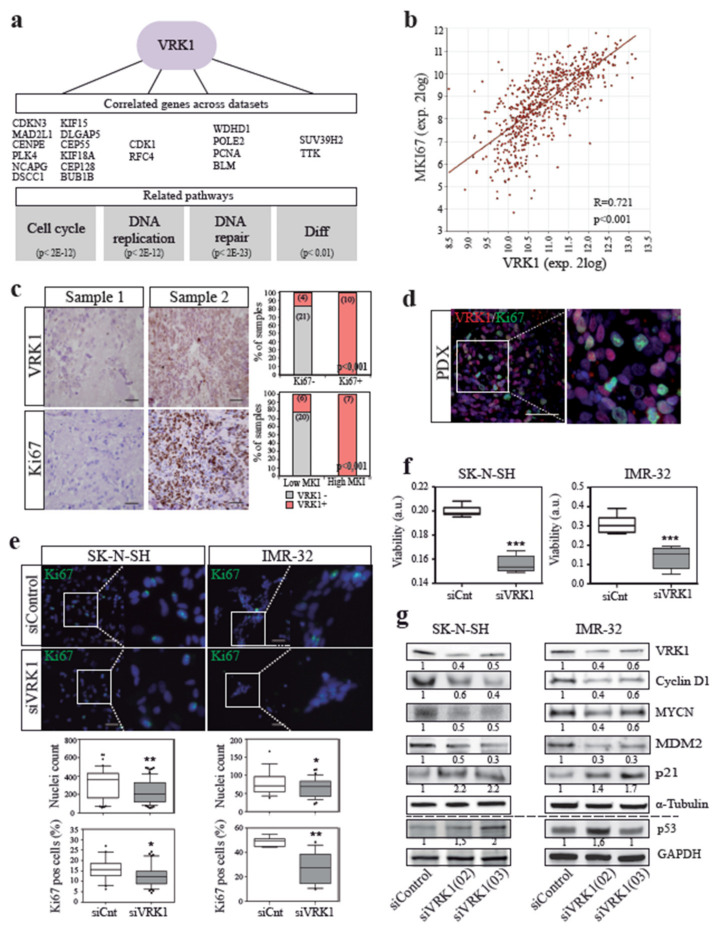
VRK1 associates with neuroblastoma tumor cell proliferation. (**a**) Representative genes correlated with VRK1 expression across NB sample datasets and their associated pathways. p values for signaling Gene Expression Omnibus (GEO) terms are shown; (**b**), mRNA expression correlation between VRK1 and Ki67 in human NB tumors (GSE45547); (**c**) immunohistochemical detection of the proliferation marker Ki67 and VRK1 in serial sections of NB patient samples. Representative negative (left) and positive (right) cases have been selected. Quantifications show correlation between VRK1 protein expression and Ki67 expression or mitotic-karyorrhexis index (MKI) per sample. Number of samples per group shown in brackets; (**d**) immunofluorescence staining showing a VRK1 and Ki67 cellular staining in a PDX tumor sample. Scale bar: 250 μM; (**e**) immunofluorescence staining and quantification of Ki67-positive cells in NB cell lines after 96 h transfection with either VRK1 or control siRNAs. Nuclei are counterstained with DAPI. Scale bar: 100 μM. ** *p* < 0.01; * *p* < 0.05; (**f**) Viability of NB cells after VRK1 downregulation by siRNA. *** *p* < 0.001; (**g**) Western blot analysis showing the levels of the indicated proteins after transient VRK1 downregulation. Quantification of representative blots shown. Note average 50% reduction in VRK1 expression.

**Figure 4 cancers-12-03465-f004:**
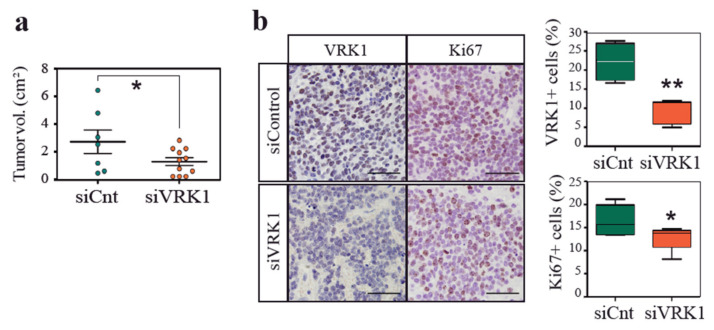
VRK1 downregulation impairs neuroblastoma tumorigenesis in a xenograft model. (**a**) Volume of tumors in mice collected after flank injection with SK-N-SH cells 24 h after transfection with either VRK1 or control siRNA; (**b**) Immunohistochemical staining showing VRK1 and Ki67 protein expression in representative tumors from each condition. Percentage of positive cells are shown. Scale bars: 60 μM.

**Figure 5 cancers-12-03465-f005:**
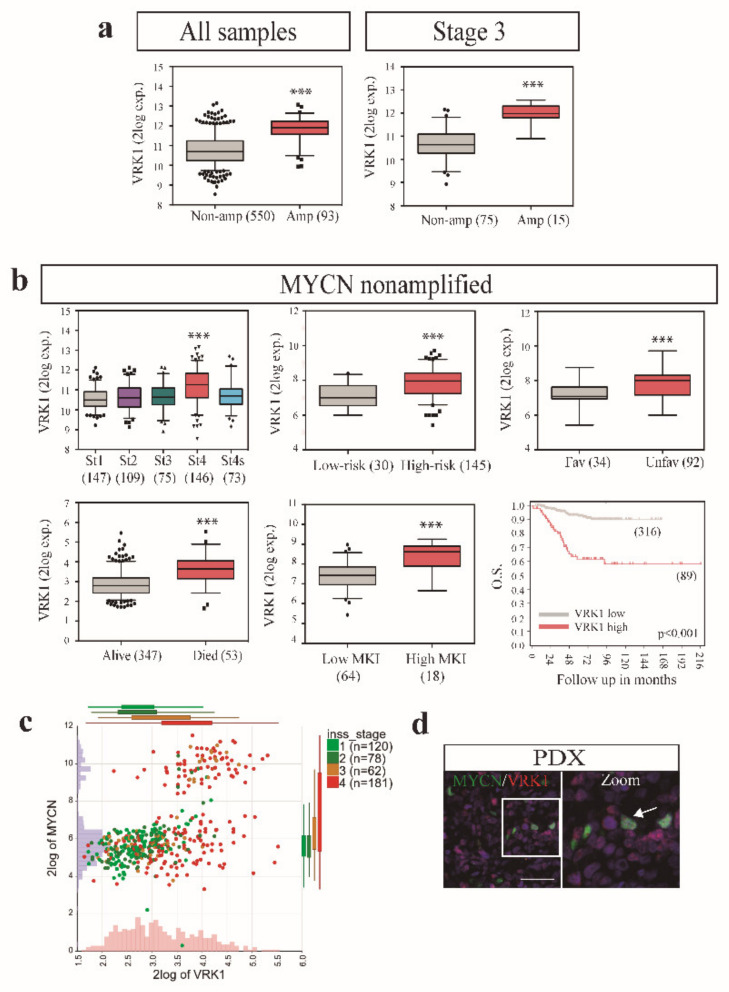
VRK1 associates with neuroblastoma progression independently of MYCN amplification. (**a**) Expression of VRK1 in MYCN-amplified or MYCN nonamplified NB tumors (GSE45547); (**b**) VRK1 expression on patient tumor samples without MYCN amplification separated by INSS stage (GSE45547), risk, histology, outcome (GSE62564) or mitotic-karyorrhexis index (MKI) (GSE3446). Kaplan curves show overall survival probability from patients stratified according to VRK1 expression (GSE45547). Number of samples shown in brackets. ***: *p* < 0.001; (**c**) two gene (VRK1 and MYCN) expression analyses in tumor samples (dataset GSE49710). Number of samples in each INSS stage are shown and separated by colors; (**d**) immunofluorescence showing MYCN and VRK1 staining in a representative PDX sample.

**Figure 6 cancers-12-03465-f006:**
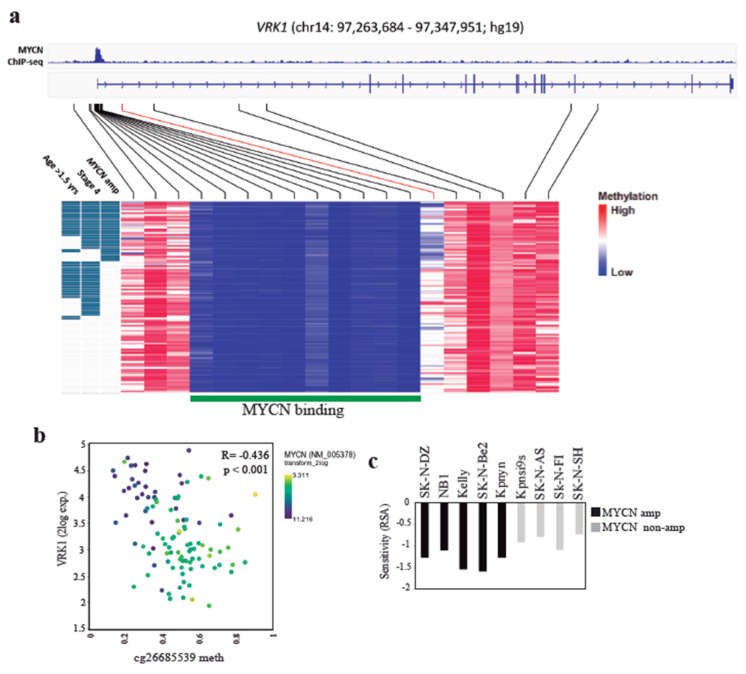
VRK1 promoter is hypomethylated in MYCN-amplified NB tumors. (**a**) Graphic representation of the VRK1 gene promoter region and colour-coded methylation levels on 16 regional CpG probes across 105 NB tumor samples. The MYCN binding site according to MYCN ChIP-seq in neuroblastoma cell lines Be(2)-C, Kelly and NGP is depicted in green with exemplary Be(2)-C MYCN ChIP-seq signal at the top. cg2668539 probe is shown in red; (**b**) Correlation between VRK1 expression and the methylation of the probe cg26685539 in NB tumors. Each dot represents a patient sample and is colored code according to MYCN gene expression; (**c**) MYCN status-dependent sensitivity (RSA) for VRK1 knockdown in 9 NB cell lines. Negative RSA values indicate viability reduction upon knockdown.

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
