# Peer review of "Identification of VRK1 as a New Neuroblastoma Tumor Progression Marker Regulating Cell Proliferation"

_cancers, 2020, doi:10.3390/cancers12113465_

Round 1
Reviewer 1 Report
The manuscript “Identification of VRK1 as a new neuroblastoma tumor progression marker regulating cell proliferation” by Ana etal is nicely written. The study is of great potential for the readers of the journal. Major concern regarding this study is about the prolonged effect of VRK1 siRNA on tumor growth. Did authors check the expression of VRK1 in the VRK1 depleted tumors? The authors should discuss what the probable mechanism of maintenance of VRK knockdown in tumor microenvironment.
Author Response
We thank the reviewer for his/her comments. Regarding the expression of VRK1 in VRK1 depleted tumors, we checked the expression of this and other proteins by IHC, both in xenografts derived from VRK1 knockdown cells and controls. As shown in Figure 4b, VRK1 expression was consistently lower in tumors derived from VRK1 depleted cells. We still do not have a clear explanation regarding the mechanism responsible for the maintenance of VRK1 knockdown in tumors. We have discussed a possible epigenetic effect affecting VRK1 gene downregulation on our manuscript (line271). In our hands, neuroblastoma tumor xenografts need to surpass a threshold in the number of tumor cells to successfully grow in mice. Following cell injection, tumors take a considerable time to develop, but tumor growth is very fast once it reaches approximately 10mm3. It is possible that VRK1 depletion makes reaching a critical number of cells in the tumor harder. These cells, not entering a proliferative state, could keep VRK1 low for longer. This case would raise the possibility of paracrine or tumor microenvironment insults onto proliferating NB tumor cells, affecting VRK1 gene expression and proliferation. We have introduced this possibility also in the discussion (line 276).
Reviewer 2 Report
This is a highly exciting and important paper with several unexpected problem, which makes this paper contradictory. The presence of a real clinician with experience in the field of pediatric oncology would help for authors to avoid several basic problems.
Major comments:
The risk stratification is highly confused. The authors concomitantly use INSS (stage 1-4) and INRG (very low-low-intermediate - high risk) and INRGSS neuroblastoma stratification systems, which highly disturbs the results section and Figures, makes the interpretations of their finding confusing.
Minor comments:
Abstract is too long (250 words) highly surpasses permitted maximum 200 words.
Mentioning of INRG system occurs much later in the text, when this stratification is used (line 126 or 367 vs Fig 1)
INRG or INSS or INRGSS systems should be quoted in Reference section.
Abbreviations should be solved (eg PDX)
First paragraph of the Result section left in the text from Intructions for Authors
Line 135: authors set SK-N-S against IMR-32 in experiment many times to prove something. However, it is never stated in the manuscript what is the main difference between these two cell lines, what is the aim of this setting of experiments.
Fig 3e. Definition of VRK1-1 and VRK1-2 is nor in the text nor in Fig Legend.
Author Response
Response to major comments:
We are sorry if the use of sample cohorts initially diagnosed following different staging systems was not clearly described in our manuscript. Firstly, we have made use of publicly available expression datasets with patient samples classified at diagnosis following the INSS system (stages 1 to 4). These are described in the section subheading 4.1 Human tumor gene expression and methylation analysis of Materials and Methods section. In some cases, the clinical information available for these samples included the Risk stratification information, using the INRG, and this has been used to analyse VRK1 expression in the cohorts. We have further clarified this matter in the methods adding information in line 312.
In addition, for immunohistochemistry, we have used our own patient sample cohort, more recent and classified initially following the INRG staging system (INRGSS; stages L1-2, M). These are described in the subheading 4.5 Immunofluorescence and immunohistochemistry of the Materials & Methods section. The sets of patient samples classified according to different staging systems have not been mixed in the analysis, and the patient cohort used has been described in the figure legends.
Minor comments:
Abstract is too long (250 words) highly surpasses permitted maximum 200 words.
(R) We have now edited the abstract to shorten it.
Mentioning of INRG system occurs much later in the text, when this stratification is used (line 126 or 367 vs Fig 1)
(R)We have now corrected figure legends and text to correct this.
INRG or INSS or INRGSS systems should be quoted in Reference section.
(R) Original references to the different classification systems have been added when described in the Methods section.
Abbreviations should be solved (eg PDX)
(R) We have now corrected old abbreviations that were not cited correctly on first appearance.
First paragraph of the Result section left in the text from Intructions for Authors
(R) We are sorry for this mistake we made while preparing our manuscript. Paragraph has been now eliminated.
Line 135: authors set SK-N-S against IMR-32 in experiment many times to prove something. However, it is never stated in the manuscript what is the main difference between these two cell lines, what is the aim of this setting of experiments.
(R) Initially we picked these two cell lines to make our findings more robust, not necessarily expecting different phenotypes regarding VRK1, but to confirm our results in independent cell lines. These cell lines are however quite different and can be taken as paradigm of different entities in NB tumor cells. SK-N-SH are MYCN non amplified, and IMR-32 has amplification of MYCN. Moreover, we and others have described a phenotypic heterogeneity in SK-N-SH absent in IMR-32, which is a typical adrenergic cell line (Vega et al. EBioMedicine 49 (2019) 82–95; van Groningen T et al. Nat Genet 2017;49:1261–6). We believe that the fact that we observe a similar phenotype in both, makes our findings of more general application.
Fig 3e. Definition of VRK1-1 and VRK1-2 is nor in the text nor in Fig Legend.
(R) We guess reviewer refers to labelling in Fig. 3g. The reviewer is right and this is a mistake, as we have used siRNAs with names siVRK1(02) and siVRK1(03), and we describe them as such in the Methods section. Figure 3g has been corrected.
Reviewer 3 Report
The paper of Colmonero-Repiso et al deals with the potential synergic action of Vaccinia-related kinase 1 (VRK1) and oncogene MYCN in neuroblastoma progression.
The purpose of the work is well presented. Currently, no evidence has been reported that directly links VRK1 and neuroblastoma and the rationale to investigate the connection with MYCN is a good point.
The correlation between increased expression and protein detection of VRK1 in neuroblastoma tumor samples with the relapses and worse outcome is convincing.
The suggested functional role is convincing but also other biochemical pathways are possible. On the whole the paper give interesting insights to understand the progression of neuroblastoma and suggest new targets for the treatment of this tumor. The paper deserves publication in Cancers
Author Response
(R) We thank the reviewer for his/her positive comments and the consideration about our results.
Reviewer 4 Report
cccccccccccccccccccccccccccccccccccccccccccccccccccc
General comment. The paper is well written, complete, ad easy to read even for a clinician, the analyzes carried out very detailed. The figures are very understandable as well as the explanations. The structure of the paper is correct and the reading is fluid.
Minor comments. The authors describe VRK1 as a new potential marker in NB able to predict the final prognosis independently of already well known factors such as MYCN status, histology, etc., in particular this new factor would be able to better discriminate the prognosis in NB “intermediate risk”.
It is well known that age is one of the most important prognostic factor in NB in both localized or metastatic disease. Did you correlate the expression of VRK1 with the age of patients at diagnosis? In particular in the age group : 1) < 12 mos, 2) 12-18 mos, 3) > 18 mos. Does the predictive value of VRK1 remain unchanged?
For example in patients with metastatic NB but < 12 mos at diagnosis the role of MYCN (rarely amplified) , histology or chromosomal abnormalities is not so clear while it would be very important to find a new prognostic marker !. Similarly in intermediate risk NB particularly in patients with NB L2 (according INRG) >18 mos whose final prognosis is not yet optimal (LINES-SIOPEN trial in progress) it is important to study and validate new markers.
From a statistical point of view, is the cohort of MYCN- stage 3 patients numerically adequate to draw final conclusions about the role of VRK1?
Author Response
Minor comments. The authors describe VRK1 as a new potential marker in NB able to predict the final prognosis independently of already well known factors such as MYCN status, histology, etc., in particular this new factor would be able to better discriminate the prognosis in NB “intermediate risk”. It is well known that age is one of the most important prognostic factor in NB in both localized or metastatic disease. Did you correlate the expression of VRK1 with the age of patients at diagnosis? In particular in the age group : 1) < 12 mos, 2) 12-18 mos, 3) > 18 mos. Does the predictive value of VRK1 remain unchanged?
(R) We are thankful to the reviewer for these very useful comments. We have now checked VRK1 expression and stratification of patients in a patient cohort dividing them by the age groups suggested. The expression of VRK1 is not overall significantly different between the different age groups. However, VRK1 expression is statistically different among patients with MYCN non amplified tumors, being higher in the >18 months group. Interestingly, VRK1 expression alone is able to stratify the patients in these age groups with worse survival. A high VRK1 expression is indicative of low survival in every age group, both considering patients with or without MYCN amplification, although the number of patients in the groups formed in some cases is too low to reach a definitive conclusion. We have now discussed this (line 120) and added some data showing this in new Supplementary Figure S1c.
For example in patients with metastatic NB but < 12 mos at diagnosis the role of MYCN (rarely amplified) , histology or chromosomal abnormalities is not so clear while it would be very important to find a new prognostic marker !. Similarly in intermediate risk NB particularly in patients with NB L2 (according INRG) >18 mos whose final prognosis is not yet optimal (LINES-SIOPEN trial in progress) it is important to study and validate new markers. Regarding patients >18 months at diagnosis, VRK1 expression stratify patients with low survival among those, and among intermediate stage patients.
(R) Unfortunately, we do not have access to patient cohorts with enough number of samples and classified at diagnostic following the INRG staging system to be able to perform detail studies in the indicated groups. We have seen significant stratification of patients with worse survival according to VRK1 expression in patients <12 months, even in samples without MYCN amplification.
From a statistical point of view, is the cohort of MYCN- stage 3 patients numerically adequate to draw final conclusions about the role of VRK1?
(R) We understand the concern of the reviewer and would have like to be able to use a bigger cohort. However, Stage 3 MYCN non amplified tumors is a minority subgroup and, among them, only few of the patients with these tumors present low survival. We have not been able to analyse a bigger cohort but are confident with the differences we have seen on survival among these patients when stratified according to VRK1 expression. We have added a comment clarifying this point in the discussion (lines 288-290)
Reviewer 5 Report
Minor
- Lines 89-91. These are editing comments pertaining to the format of the manuscript, not the scientific content of the study. Please, delete from the manuscript.
- The introduction should describe the family of chromatin remodelling enzymes pertaining to VRK1, with emphasis on epigenetic regulators relevant to neuroblastoma biology.
- Line 119. “show” should be “shows”.
- Figure 2. The authors claim that mRNA and protein levels of VRK1 correlate (Line 118). However, the protein data presented in Figure 2 is only qualitative and claims of correlation require statistical quantification.
- Figure 3g shows that VRK1 knockdown results in a decrease in MDM2 levels and an increase in P21. This observation indicates an increase in TP53 activity. This finding should be discussed in the light of predominant wild-type TP53 activity in neuroblastoma and sensitivity of MYCN-amplified neuroblastoma tumours to nutlin (Petroni M et al. MYCN sensitizes human neuroblastoma to apoptosis by HIPK2 activation through a DNA damage response. Mol Cancer Res. 2011;9(1):67-77; Chen et al. Pre-clinical evaluation of the MDM2-p53 antagonist RG7388 alone and in combination with chemotherapy in neuroblastoma. Oncotarget. 2015;6(12):10207-21).
- The authors argue in Figure 4 that VRK1 knockdown slows tumorigenesis. However, tumorigenesis is the process by which a normal cell transitions to a malignant states and is allowed to grow a tumour through signalling with the surrounding environment as well as a result of immunoediting. The transplant model in immunocompromised mice does not recapitulate neuroblastoma tumorigenesis. Thus, the interpretation of this experiment should be modified. In my opinion, Figure 4 belongs to an additional panel of Figure 3, which shows reduced proliferation following VRK1 knockdown. The observation of the long-lasting effects of VRK1 knockdown in vivo improve the strength of the manuscript.
Major:
- Section 4.1. Human tumor gene expression and methylation analysis. Figure 1 shows strong evidence of the prognostic value of VRK1. Stratifying biomarkers are only useful if they can predict the outcome. Therefore, only VRK1 expression prior treatment should be correlated with outcome. Please provide information as to the timing of biopsy relative to treatment. Clinical characteristics of the human patient cohorts need to be provided. For instance, did the patient receive neo-adjuvant chemotherapy prior biopsy? This would influence analysis of VRK1 expression.
Author Response
Minor
- Lines 89-91. These are editing comments pertaining to the format of the manuscript, not the scientific content of the study. Please, delete from the manuscript.
(R) We are sorry for this mistake we made while preparing our manuscript. Paragraph has been now eliminated.
2. The introduction should describe the family of chromatin remodelling enzymes pertaining to VRK1, with emphasis on epigenetic regulators relevant to neuroblastoma biology.
(R) Thank you for the suggestion. We have now added some comments and references regarding the chromatin remodelling activity of VRK1. It phosphorylates several histone subunits affecting acetylation and methylation and regulating gene transcription. This has been better documented in the context of DNA damage response and cell cycle, although the complete transcriptional profile affected by VRK1 has not been explored. To the best of our knowledge, no data regarding the relationship between VRK1 and other epigenetic regulators in neuroblastoma has been published. There is a report in Ewing Sarcoma describing how the EWS-FLI1 transcription factor remodels chromatin, affecting in turn VRK1 gene (Riggi et al. Cancer cell, 2014)
3. Line 119. “show” should be “shows”.
(R) Mistake has been corrected.
4. Figure 2. The authors claim that mRNA and protein levels of VRK1 correlate (Line 118). However, the protein data presented in Figure 2 is only qualitative and claims of correlation require statistical quantification.
(R) We agree that “Correlation” is the wrong word to be used in the text on this sentence as it is prone to misunderstanding. We have changed the wording to reflect that we have analyse VRK1 protein expression, in addition to mRNA (line 124).
5. Figure 3g shows that VRK1 knockdown results in a decrease in MDM2 levels and an increase in P21. This observation indicates an increase in TP53 activity. This finding should be discussed in the light of predominant wild-type TP53 activity in neuroblastoma and sensitivity of MYCN-amplified neuroblastoma tumours to nutlin (Petroni M et al. MYCN sensitizes human neuroblastoma to apoptosis by HIPK2 activation through a DNA damage response. Mol Cancer Res. 2011;9(1):67-77; Chen et al. Pre-clinical evaluation of the MDM2-p53 antagonist RG7388 alone and in combination with chemotherapy in neuroblastoma. Oncotarget. 2015;6(12):10207-21).
(R) We thank the reviewer for these comments and insights. TP53 is indeed activated in NB cells after VRK1 knockdown, although the results on IMR-32 are not entirely consistent. Therefore, VRK1 inhibition could impair NB tumor growth partially by the activation of wild-type p53. We have now added this information to Figure 3g.
6. The authors argue in Figure 4 that VRK1 knockdown slows tumorigenesis. However, tumorigenesis is the process by which a normal cell transitions to a malignant states and is allowed to grow a tumour through signalling with the surrounding environment as well as a result of immunoediting. The transplant model in immunocompromised mice does not recapitulate neuroblastoma tumorigenesis. Thus, the interpretation of this experiment should be modified. In my opinion, Figure 4 belongs to an additional panel of Figure 3, which shows reduced proliferation following VRK1 knockdown. The observation of the long-lasting effects of VRK1 knockdown in vivo improve the strength of the manuscript.
(R) We thank the reviewer for raising this point and agree with the suggestion. We have modified the text related to Fig. 4 to change “tumorigenesis” for “tumor progression” (line 189). We also agree that Figure 4 could be an extra panel of Figure 3, however we had decided to place it in a different figure for clarity and editing purposes, as Figure 3 already had quite a few panels. We have decided to leave these results as Fig. 4 pending on decision and recommendation by the editorial team.
Major:
- Section 4.1. Human tumor gene expression and methylation analysis. Figure 1 shows strong evidence of the prognostic value of VRK1. Stratifying biomarkers are only useful if they can predict the outcome. Therefore, only VRK1 expression prior treatment should be correlated with outcome. Please provide information as to the timing of biopsy relative to treatment. Clinical characteristics of the human patient cohorts need to be provided. For instance, did the patient receive neo-adjuvant chemotherapy prior biopsy? This would influence analysis of VRK1 expression.
(R) We thank the reviewer for this comment, which is relevant. Clinical characteristics of the patient cohorts from publicly available expression data can be accessed on R2, following the accession numbers in Supplementary Table S1. Patient sample cohort used for stratification of survival data on Figure 1, according to the information available, is from biopsies at diagnostic before treatment and have been used for prediction elsewhere. Our results on expression of VRK1 on pre- and post-treatment samples by IHC (Supplementary Fig. 1), indicates indeed that treatment could affect the expression of VRK1.
Round 2
Reviewer 1 Report
Authors have addressed all the comments. The manuscript can be considered for publication.
Reviewer 2 Report
It is nice and important paper especially for pediatric oncoloigists, which is ready for publication in Cancers in its present form
Reviewer 5 Report
Dear authors,
Thank you for the amendments and congratulations on your manuscript.
All the best,